# Conserved sequence motifs in the abiotic stress response protein late embryogenesis abundant 3

Karamjeet K. Singh[1], Steffen P. Graether[1,2]*

1 Department of Molecular and Cellular Biology, University of Guelph, Guelph, Ontario, Canada, 2 Graduate Program in Bioinformatics, University of Guelph, Guelph, Ontario, Canada

* graether@uoguelph.ca

## Abstract

LEA3 proteins, a family of abiotic stress proteins, are defined by the presence of a trypto-phan-containing motif, which we name the W-motif. We use Pfam LEA3 sequences to search the Phytozome database to create a W-motif definition and a LEA3 sequence dataset. A comprehensive analysis of these sequences revealed four N-terminal motifs, as well as two previously undiscovered C-terminal motifs that contain conserved acidic and hydrophobic residues. The general architecture of the LEA3 sequences consisted of an N-terminal motif with a potential mitochondrial transport signal and the twin-arginine motif cut-site, followed by a W-motif and often a C-terminal motif. Analysis of species distribution of the motifs showed that one architecture was found exclusively in Commelinids, while two were distributed fairly evenly over all species. The physiochemical properties of the different architectures showed clustering in a relatively narrow range compared to the previously studied dehydrins. The evolutionary analysis revealed that the different sequences grouped into clades based on architecture, and that there appear to be at least two distinct groups of LEA3 proteins based on their architectures and physiochemical properties. The presence of LEA3 proteins in non-vascular plants but their absence in algae suggests that LEA3 may have arisen in the evolution of land plants.

## Introduction

Plants are often subjected to a variety of abiotic stresses that can restrict their growth and potentially result in death, where drought and cold stresses are thought to have the most significant effects on crop growth [1]. Both of these stresses lead to dehydration at the cellular and whole-plant level, causing a decrease in photosynthetic reaction rates and an increase in the production of reactive oxygen species (ROS). ROS do have roles in cell signaling and homeostasis; however, over-accumulation of ROS can lead to oxidative stress, thereby damaging or impairing the function of DNA, proteins, and lipids [2].

As sessile organisms, plants have evolved to respond to adverse environmental conditions using a number of different adaptations. They can respond to dehydration by modifying their

**Data Availability Statement:** All relevant data are within the manuscript and its Supporting Information files.

**Funding:** This work was supported by a National Research and Engineering Council (Canada) grant

to SPG and an Ontario Graduate Scholarship to KKS.

**Competing interests:** The authors have declared that no competing interests exist.

root architecture to create a deep and thick root system to enhance their ability to capture soil moisture, and by closing stomata and reducing leaf surface area to minimize water loss [3]. There is also an overproduction of various osmolytes (e.g., sugars, sugar alcohols, small dipeptides, amino acids) to help regulate water levels, minimize ROS formation, and stabilize enzymes, as well as the synthesis of antifreeze proteins to prevent the formation of ice and/or inhibit ice crystallization [4–6]. Also, there is an upregulation in the expression of late embryogenesis abundant (LEA) proteins, which have been shown to confer dehydration and cold tolerance to plants [7–9].

LEA genes appear to be numerous in the plant genome, with 51 *lea* genes identified in *Arabidopsis thaliana* [10, 11]. As their name suggests, LEA proteins accumulate in seeds during the later stages of embryogenesis, but are also expressed in all plant life stages. In adult plants, there is an up-regulation of the genes encoding for LEA proteins in vegetative tissues after exposure to dehydrative, low temperature, and/or osmotic stresses [10, 11]. LEA proteins possess a biased amino acid composition that is rich in glycine and other small and/or charged amino acids, while containing a minimal number of cysteines, non-polar and aromatic amino acids. The prevalence of hydrophilic residues favors the association of LEA proteins with water, resulting in an open, random coil structure in solution. Not surprisingly, LEA proteins are classified as intrinsically disordered proteins (IDPs), meaning they lack stable secondary and tertiary structure [12, 13].

The LEA proteins found in the model plant *A. thaliana* were grouped based on sequence similarity, although the naming convention and grouping remains inconsistent in the literature and sequence databases. Here, the Pfam naming system will be used, as detailed by Hundertmark and Hincha [11].

Many plant species possess multiple LEA proteins, and the expression of LEA proteins has been in observed in many intracellular compartments, including the cytosol [14, 15], chloroplasts [16, 17], endoplasmic reticulum [18], peroxisomes [19], nuclei [20, 21], and mitochondria [22–24]. Group 2 LEA proteins, known as dehydrins, are the best characterized of all the LEA groups [25, 26]. Group 3 is also of interest due to localization to the mitochondrion [27], an important organelle for energy production.

Currently, very little specific data exist for the *A. thaliana* group 3 LEA proteins; however, one member, SAG21 (also known as LEA3-2), has been studied. Transgenic plants that overexpressed AtLEA3-2 showed a higher root and shoot biomass, under both normal growth conditions and in the presence of $H_2O_2$ [28]. They also showed that LEA3-2 is upregulated in response to oxidative- and drought-induced stress, and this up-regulation is beneficial to plant growth and survival. A follow-up study further explored the potential function of LEA3 proteins by using both overexpression and anti-sense *LEA3-2* plant lines [29]. While overexpression produced taller plants with more flowering stalks, the anti-sense plants were shorter, had fewer flowering stalks, fewer leaves, and lower rosette biomass. They also showed that the protein localized to the mitochondrion [29]. These studies suggest that LEA3-2 may play a role in the function/stability of mitochondrial proteins involved in ROS production or signaling [28, 29]; however, the mechanism of action at the cellular level and biochemical characterization still requires further studies.

Genome-wide analysis of LEA proteins in a single species have been previously performed to evaluate common motifs, expression patterns, evolution, and predicted localization [11, 30–35] Multi-genome analyses on the dehydrin group resulted in a more rigorous and consistent motif description [36] and provided insight into their evolution [37]. However, to date there is a lack of targeted studies investigating other LEA groups, such as LEA3, across multiple plant species. To aid in furthering our understanding of LEA3 proteins, we perform multiple bioinformatics analyses in here to rigorously define the conserved motifs and architectures in

vascular and non-vascular plants, and examine how LEA3 proteins are spread throughout plant species.

## Materials and methods

### LEA3 protein motifs

The goal was to first find a plausible W-motif that could be used to perform a more exhaustive search to find LEA3 protein sequences in a large number of plant species. The initial search for LEA3 protein sequences was performed using the Pfam PF03242 NCBI sequences as the query sequences for a BLAST search against all protein sequences in the Phytozome v13 (primary transcripts) with an $E$-value cut-off of $10^{-6}$. A list of the higher plant genomes that were searched are listed in S1 Table. Sequences with $\geq$99% identity were trimmed from the results list such that only one sequence example remained. To develop a more comprehensive tryptophan containing motif for the next search, MEME [38] was run on those sequences using the "any number of repeats" mode, searching for 10 sites with a maxsites value of 3000. All other parameters were left at their default values.

The resulting W-motif sequence was used as the search query with FIMO [39] against all Phytozome v13 protein sequences from vascular plants (named higher plants, listed in S1 Table), using a threshold cut-off of $10^{-7}$, with all other settings at their default values. This threshold value was determined empirically by scanning results with cut-off values between $10^{-5}$ and $10^{-12}$. The $10^{-7}$ cut-off was chosen on the basis that the W-motif could be detected in the MEME search, but would still include sequences that may not have been detected by the initial BLAST search. This resulting sequence dataset (S1 Appendix) was used for all subsequent analyses.

MEME was run on the LEA3 protein sequence dataset to obtain a representative W-motif of LEA3 proteins, and to discover other motifs. Optimal motif widths were determined by varying the widths by ±2 residues based on the initial MEME run, and then keeping the motifs with a higher number of positions with conserved ($\geq$67%) positions either by amino acid or by same physiochemical property (charged, hydrophobic, polar or aromatic). Motifs that occurred in >20% of all sequences were further inspected for inclusion as a conserved motif. All searches were performed with the "any number of motifs" mode, with the top 10 motifs returned. Motifs were visualized using the LOGO representation [40]

We separately searched for LEA3 proteins in the genomes of lower plants and green algae. A list of the lower plant species, primarily defined as non-vascular plants, is included in S1 Table. Exceptions to the non-vascular species is *Selaginella moellendorfii*, which was included as a lower plant because it is the oldest extant species among tracheophytes, and the gymnosperms *Ginkgo biloba* and *Picea abies*, due to their evolution long before angiosperms. A list of the algae species examined is included in S1 Table. For lower plants, a FIMO search was performed using the W-motif definition from higher plants to create the lower plant sequence dataset with a threshold of $10^{-9}$. A FIMO search (threshold of $10^{-4}$) using the RRGYA$_4$ motif and a BLAST search ($E$-value 0.01) using LEA3 protein sequences from higher plants did not find any additional hits. Subsequent analyses (motif discovery) were performed as described for higher plants.

For algae, BLAST searches ($E$-value of 0.1) were performed using both higher and lower plant LEA3 protein sequences. Motif searches were performed with FIMO (threshold of 0.001) and MAST ($E$-value of 0.01) using the W-motif, DAELR motif and RRGYA$_4$ motif.

### LEA3 protein architecture and species tree

The eight motifs discovered by MEME were used as input for [41] in order to define the LEA3 architectures in higher plants. All parameters were left at their default settings. To determine

the number of residues between conserved motifs (i.e. the variable regions of the protein sequences), an in-house script was developed to analyze the MAST results. Likewise, an in-house written script was used to extract all residues not located in the conserved motifs in order to determine the amino acid composition of those regions.

A phylogenetic tree of all plant species used in this study was created using the PhyloT tree generator server (https://phylot.biobyte.de). The NCBI taxonomic reference numbers were obtained using the NCBI genome browser, and then used to infer an NCBI taxonomic identifier tree. The species were divided into clades consisting of Commelinids, Asterids, Malvids, and Fabids. Species falling outside of these groups but containing only one or two examples were not included in the analysis. To determine the fraction of one architecture within all four clades, the fraction of one architecture was first calculated by dividing the number of one architecture in one clade by the total number of LEA3 proteins in that clade. This value was then normalized by multiplying it by the ratio of the fractional number of LEA3 proteins within a clade divided by the number of species within the clade. Likewise, the fraction of an architecture within one clade was calculated by dividing the total number of LEA3 proteins with that architecture by total number of LEA3 proteins in that clade.

### LEA3 protein properties

The isoelectric point (pI), size of the protein (molecular weight), hydrophobicity (GRAVY score) and overall disorder propensity (FoldIndex) of the LEA3 proteins were calculated. Protein sequences were grouped on the basis of the N-terminal motif. Analyses were performed using the Gene Infinity Server (http://www.geneinfinity.org) for pI, MW, and GRAVY scores [42], and the FoldIndex server (https://fold.weizmann.ac.il) for the FoldIndex score [43]. The data were plotted as bean plots using the bean plot package [44] in *R* [45].

### LEA gene evolution

The sequence dataset was used to create a multiple sequence alignment (MSA) using the Multiple Sequence Comparison by Log-Expectation (MUSCLE) tool on the EMBL-EBI server [46]. The MSA was then input into ProtTest 3.4.2 [47] to determine the best-fit model of protein evolution, which suggested using the JTT+G model [48]. Next, the MSA and protein evolution model were used to search for the best maximum likelihood (ML) tree using RAxML-ng [49], performing 100 searches starting with 50 random and 50 parsimony trees. The best tree was used for 1000 bootstrap replicates. The tree was visualized using MEGA X [50].

## Results

### Conserved motifs in higher plants

The Pfam dataset (PF03242) is a collection of protein sequences that are annotated as LEA3, likely due to the presence of a conserved tryptophan-containing motif [11, 30–35]. After eliminating any duplicate sequences, the dataset was used as an input for the MEME program to search for the W-motif, which was subsequently used as a search motif for FIMO [39] against the Phytozome v13 protein, primary transcript datasets. All of these protein sequences were used in a re-run of MEME to obtain a more comprehensive version of the tryptophan-containing motif (named here as the W-motif) and other newly discovered LEA3 protein motifs. The initial analysis was performed on vascular plants found in the Phytozome database (defined as higher plants in this paper).

A LOGO representation of the W-motif is shown in Fig 1A, which demonstrates a conserved tryptophan residue in position 1 (100%). This motif, by definition, is present in all

LEA3 proteins. Other highly conserved positions (i.e., frequency >67%) include prolines at positions 3, 5, and 12, aspartate at position 4, threonine at position 7, and glycine at position 8. Position 10 is completely conserved in terms of aromatic character. The remaining positions are fairly variable in terms of amino acid type and property, although position 2 seems to be predominantly hydrophobic.

Two motifs towards the C-terminal end were discovered during the search: the DAELR and EDVMP motifs (Fig 1B and 1C). The motif names are meant to emphasize the amino acids that are enriched in the sequence, and not to capture the exact motif pattern. The DAELR motif is more common in LEA3 proteins, being detected in 75% of the sequences, while EDVMP motif is present in 20% of them. The DAELR motif has several completely conserved residues; aspartate at position 3, leucine at position 7 and arginine at position 8. The hydrophobic character is preserved at positions 2, 4, and 5, with alanine being common in position 5. Negatively charged amino acids are common at positions 1 and 6. The last three residues are variable, but often contain lysine, asparagine, arginine and glutamine (i.e., side chains with nitrogen groups). For the EDVMP motif, high conservation also exists at several positions; glutamate at position 1, valine at positions 3 and 7, methionine at position 4, aspartate at position 6, proline at position 8. Alanine is often found at positions 11 and 12.

The motif search also yielded four N-terminal motifs, which we have named as MARS, MAARS, MGRX and M[AS][RK] (Fig 1D). The MARS motif is the most common among LEA3 proteins (52%), followed by MGRX (28%). The M[AS][RK] and MAARS motifs were each found in about 10% of the sequences. The conserved regions for the MARS motif include methionine at position 1, alanine at position 2, arginine at position 3, serine at position 4 and lysine at position 9. Positions 8, 10, 11, 15 and 19 are predominantly non-polar. The MAARS motif showed largely a similar pattern other than the alanine being conserved at positions 2 and also 3. Based on this, we propose that the MAARS motif is essentially identical to the MARS motif, with the exception of the inserted alanine. Therefore, for the remainder of the paper we combine the two motifs into one, which we name the MAaRS motif, where the lowercase 'a' represents the insertion.

The MGRX motif has a conserved methionine in position 1, mostly glycine or sometimes alanine in position 2, and arginine in position 3. Positions 6 and 9 are often non-polar, and positions 11–13 often contain leucine. The M[AS][RK] motif shows considerably less conservation, with only methionine at position 1 and arginine at position 15 being >67% conserved. As suggested by the name, position 2 is mainly alanine or serine, and position 3 arginine or lysine. Positions 4, 7, 9, 11 and 12 are non-polar, and serine is often detected at positions 6, 7 and 9.

The RRGYA$_4$ motif (Fig 1E), which is found after the N-terminal motif, was observed in 67% of the sequences. Positions 1 and 2 are predominantly arginine and position 3 is often glycine. Position 4 is predominantly tyrosine, but phenylalanine is also present, suggesting aromatic character is important at this position. The final four residues are mainly alanine, with valine, threonine, and serine occurring as well.

## LEA3 protein architectures and properties in higher plants

We next examined how the various motifs are arranged in the LEA3 protein sequences, which is shown in Fig 2A, and are grouped by their N-terminal motif and by the presence and absence of the two C-terminal motifs. The first, and most common architecture, consisted of the N-terminal MAaRS motif, followed by the RRGYA$_4$ motif, a variable region that contained no identifiable motif, then the W-motif, and finally the C-terminal DAELR motif (which we denote as the MAaRS-1W architecture). The second architecture was identical to MAaRS-1W,

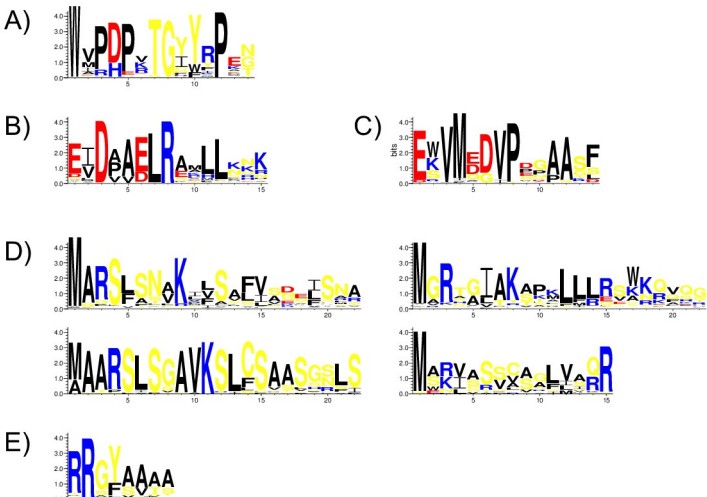

**Fig 1. Conservation of the W-motif and other motifs in LEA3 proteins.** A) W-motif. B) DAELR motif. C) EDVMP motif. D) N-terminal motifs (MARS, MAARS, MGRX and M[AS][RK]). E) RRGYA₄ motif. Amino acids are color-coded by their group type. Blue—positively charged (Lys, Arg, His); red—negatively charged (Asp, Glu); black—hydrophobic (Ala, Val, Leu, Ile, Pro, Phe, Met), yellow—polar (Gly, Ser, Thr, Tyr, Cys, Asn, Gln). The heights of the amino acids correspond to their level of conservation at that position.

except that part of the variable region contained a second W-motif (denoted MAaRS-2W). The third architecture possessed only the MAaRS motif and the W-motif, without an identifiable RRGYA₄ or DAELR motif (denoted MAaRS-no DAELR). Of these three architectures, the MAaRS-1W occurred 88% of the time, the MAaRS-2W occurred in 8% of these sequences and the MAaRS-1W-no DAELR occurred in 4%.

The remaining two architectures contained either the MGRX N-terminal motif or the M[AS][RK] N-terminal motif (Fig 2A). In both architectures, the N-terminal motif was followed by a variable region, which itself was followed by a W-motif. The M[AS][RK] architecture ended with the C-terminal DAELR motif. The MGRX architecture contained the EDVMP C-terminal motif, which was not observed in any of the other LEA3 architectures.

We subsequently analyzed the length (Fig 2B) of the variable regions of the different architectures. For sequences containing the RRGYA₄ motif, this distance was measured from the end of this motif to the beginning of the W-motif, while for the other architectures this distance was measured from the end of the N-terminal motif to beginning of the W-motif. The distance between the RRGYA₄ and the W-motif ranged from 10 to 46 residues, but the majority had a distance of 27 to 34 residues (Fig 2B, left panel). For the N-terminal to W-motif distance, a range of 10–60 residues was seen, but the range of 35–40 residues contains the highest number of sequences (Fig 2B, right panel).

Although the central region contains no identifiable motif, we examined the region to see whether it is truly variable by determining its amino acid composition (Fig 2C). Across all architectures, alanine, serine and glycine were near or over 10% in abundance, while residues that were within the 5–10% range included glutamate, lysine, arginine, threonine and valine. Amino acids that were found <1% in abundance were cysteine and tryptophan. When breaking down composition by architecture (Fig 2C), the percentages changed by small amounts while the patterns stayed for the most part the same. One notable exception is for the M[AS][RK] proteins, where the percentage for serine decreased from 13% to 7%, with the difference largely taken up by alanine increasing from 15% to 24%.

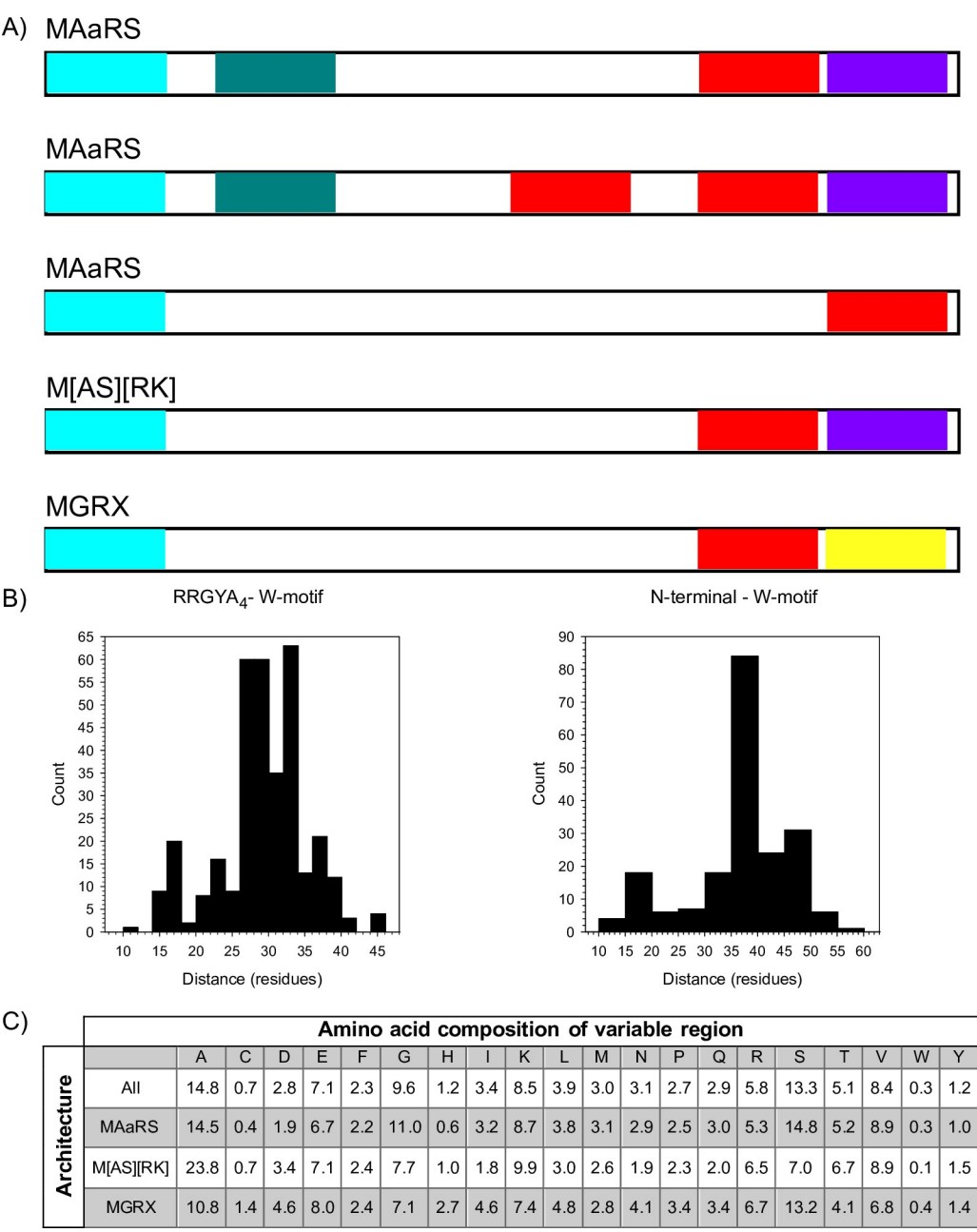

**Fig 2. Major architectures of LEA3 proteins.** A) Architecture of the LEA3 proteins grouped by the N-terminal motif and by the presence and absence of the C-terminal motifs. N-terminal motifs, light blue; RRGYA$_4$ motif, teal; W-motif, red; DAELR motif, purple; EDVMP motif, yellow. Bars are not to scale. B) Number of residues between the end of the N-terminal motif or the RRGYA$_4$ motif and the first W-motif. C) Amino acid composition of the variable region. The column headers show the single letter abbreviation of the amino acids, while the numbers represent the percent of the residues that contain that amino acid. The rows represent the composition in either all LEA3 proteins or by the major architectures.

## Distribution of LEA3 proteins among species

We next examined the distribution and number of LEA3 proteins in the different higher plant species. A full list of architectures by plant species is included as S1 Fig. Most plants (92%) have at least two LEA3 proteins, and on average there are 5±3 LEA3 proteins per

A)

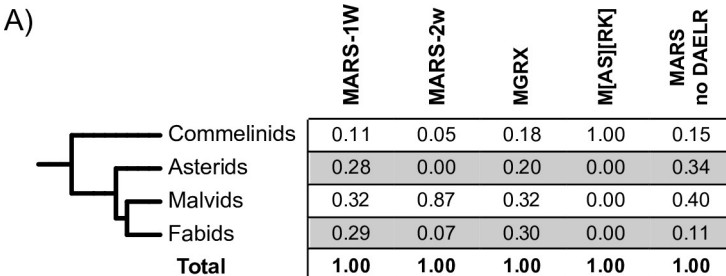

| | MARS-1W | MARS-2w | MGRX | M[AS][RK] | MARS no DAELR |
|---|---|---|---|---|---|
| Commelinids | 0.11 | 0.05 | 0.18 | 1.00 | 0.15 |
| Asterids | 0.28 | 0.00 | 0.20 | 0.00 | 0.34 |
| Malvids | 0.32 | 0.87 | 0.32 | 0.00 | 0.40 |
| Fabids | 0.29 | 0.07 | 0.30 | 0.00 | 0.11 |
| **Total** | **1.00** | **1.00** | **1.00** | **1.00** | **1.00** |

B)

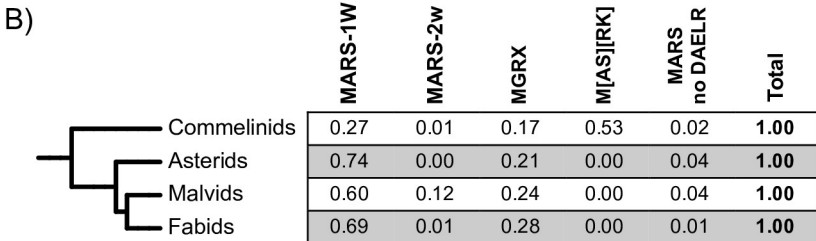

| | MARS-1W | MARS-2w | MGRX | M[AS][RK] | MARS no DAELR | Total |
|---|---|---|---|---|---|---|
| Commelinids | 0.27 | 0.01 | 0.17 | 0.53 | 0.02 | **1.00** |
| Asterids | 0.74 | 0.00 | 0.21 | 0.00 | 0.04 | **1.00** |
| Malvids | 0.60 | 0.12 | 0.24 | 0.00 | 0.04 | **1.00** |
| Fabids | 0.69 | 0.01 | 0.28 | 0.00 | 0.01 | **1.00** |

**Fig 3. Distribution of LEA3 architectures in plant clades.** The different plant species groups were combined into clades, and numbers were normalized as described in the Materials and Methods. The numbers represent A) the fraction of a plant clade that have one of the architectures, and B) the fraction of a protein architecture within one plant clade.

species. At the high end, a few species (8%) have ≥10 proteins. At the low end, *Zostera marina* has no LEA3 proteins, whereas *Glycine max*, *Mimulus guttatus*, *Oropetium thomaeum*, *Dioscorea alata*, and *Spirodela polyrhiza* have only one LEA3 protein. While examining initial analysis of the distribution LEA3 architectures among all studied species, we observed that *Arabidopsis thaliana* contained no MGRX architectures, whereas the closely related *Arabidopsis halleri* did. We therefore performed a search using FIMO and the EDVMP-motif to search for missing LEA3 proteins in *A. thaliana*, and found another LEA3 protein (AT3G19550.1) in this model plant that has not been previously identified, and we suggest that it be named AtLEA3-5.

To examine the distribution of LEA3 architectures among plants in more detail, we made a simplified grouping of plant species (Fig 3). Because these clades consist of a different number of species, all of the population fractions have been reweighted in order to normalize the results to facilitate comparison. When looking at the distribution of the five architectures across all species groups (Fig 3A), we see that MAaRS-1W and MGRX architectures were spread across the species tree. In contrast, the MAaRS-2W architecture was found predominantly among Malvids, though proteins were found throughout the species tree with the exception of Asterids, while the M[AS][RK] architecture was exclusively in the Commelinids. The absence of any C-terminal motif (MAaRS no DAELR) was more common in Malvids and Asterids, but were found in the other two groups as well.

We also examined the distribution of different architectures within each clade (Fig 3B). For all groups except for Commelinids, the MAaRS-1W architecture was the most abundant one, with the MGRX architecture being the second most common. In Commelinids, the M[AS][RK] architecture was the most abundant, while the MAaRS-1W and MGRX architectures were the next most abundant, and MAaRS-2W and MAaRS no DAELR were fairly rare.

## Physiochemical properties of LEA3 proteins

The three N-terminal motifs (MAaRS, MGRX and M[AS][RK]) were used to divide the LEA3 sequences into three groups for the analysis of their physiochemical properties. The properties that were analyzed include: isoelectric point (pI, a measure of net charge), molecular weight (MW, a measure of size), grand average of hydropathy (GRAVY score, i.e. a measure of net hydrophobicity or hydrophilicity) and propensity of a protein to fold (FoldIndex score).

The distribution of pI scores (Fig 4A) shows that MAaRS and M[AS][RK] LEA3 proteins have mainly basic pI values that are centered at pH 9.6 and 9.1, respectively, and that the majority of their sequences have a pI value between pH 9.0–11.0. There are very few acidic pI proteins in this group, with M[AS][RK] proteins being slightly more so than MAaRS. The MGRX pI values are distributed over a wider range of values (pH 5.0–9.5), with the average being near the pI value of 7.0. The MGRX protein did show a weakly bimodal distribution, with a large number of proteins having a pI centered near 6, and a smaller number having a pI centered around 9.

The molecular weight plot (Fig 4B) show that MAaRS and M[AS][RK] proteins both have molecular weights that cluster around 9–11 kDa, having an average molecular weight of about 10.5 kDa. The MGRX proteins again shows a bimodal distribution, with the molecular weight centered at 11 and 13 kDa.

Next, the GRAVY scores for the different protein groups were calculated (Fig 4C), where values greater than zero are an indicator of hydrophobicity, and values less than zero are an indicator of hydrophilicity. All LEA3 sequences show a fairly large range of values (between -1.0 and 0.0), showing that all LEA3 proteins are fairly hydrophilic. The MAaRS and M[AS][RK] sequences have similar averages of -0.4 and -0.3, while the MGRX sequences were slightly more hydrophilic, with an average value of -0.6.

Lastly, the FoldIndex score (Fig 4D) was used to assess the propensity of the LEA3 proteins to adopt a fold, with scores greater than zero indicating a high propensity to fold and scores less than zero indicating that the protein is unlikely to fold, and therefore likely be intrinsically disordered. Not surprisingly, the FoldIndex results followed a similar pattern to the GRAVY scores, with the moderately hydrophilic MAaRS and M[AS][RK] proteins having higher FoldIndex scores, with averages of 0.07 and 0.09, respectively, and the more hydrophobic MGRX proteins having a slightly lower FoldIndex scores (average of 0.03). The MAaRS proteins showed the greatest variability in hydrophobicity with values ranging from -0.16 to +0.23, while M[AS][RK] was confined to positive values between 0 to 0.5 and MGRX from -0.1 to +0.1.

## Analysis of LEA3 proteins in lower plants

We also performed similar analyses of LEA3 protein sequences from lower plants (as defined in Materials and Methods), firstly to see if they are present, and if so, how the conserved motifs and physiochemical properties may have changed over a long evolutionary time period. From the MEME analysis of lower plants, no common N-terminal motif was found among all species, though a few proteins had sequences that showed some similarity to the N-terminal motifs found in higher plants. Likewise, no C-terminal motif from higher plants (DAELR or EDVMP) motifs, nor any novel C-terminal motifs, were discovered in lower plants. The lower plant W-motif was very similar to that of higher plants, with a few small differences (Fig 5A). First, 50% of the lower plant sequences had two tryptophan residues to start the motif. Position 11 in lower plants, which in higher plants is position 10 and predominantly tyrosine, showed a more even distribution among aromatic residues in lower plants (Fig 5A). Lastly, position 14 seemed to have a highly conserved glutamate residue. A RRGYA$_4$ motif was also detected,

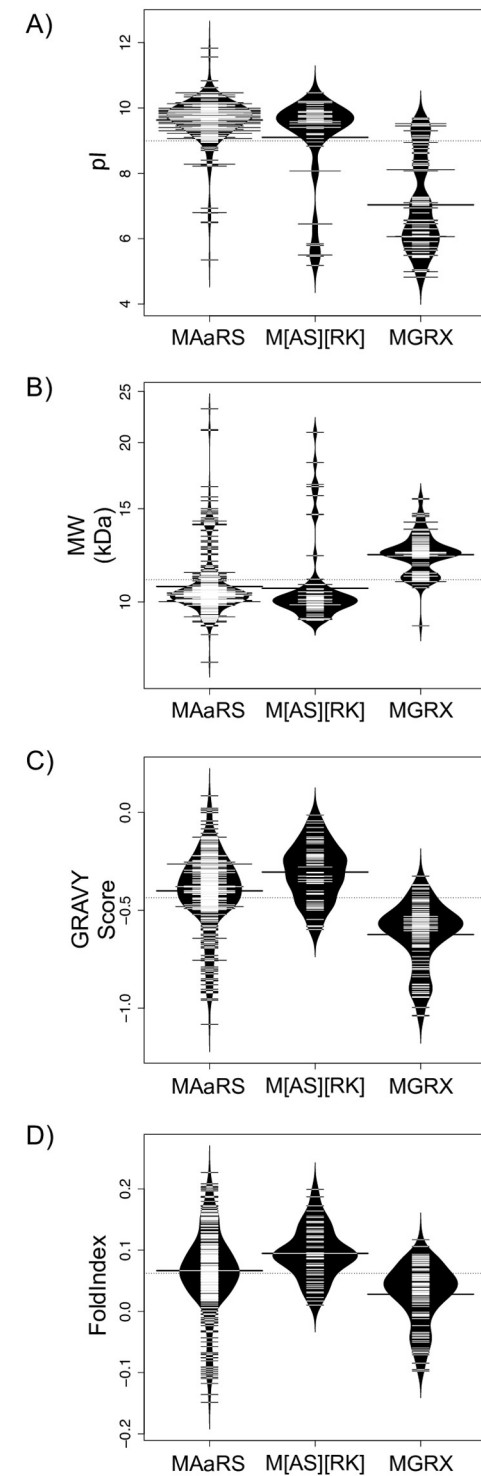

**Fig 4. Physiochemical properties of LEA3 proteins.** Bean plots of the A) isoelectric point (pI), B) molecular weight (MW), C) GRAVY score and D) FoldIndex score of LEA3 proteins grouped by the three major N-terminal architectures. The thin bars show the value of an individual protein, the wider black bar shows the mean value of an architecture, and the dotted line shows the mean value of all protein sequences. The violin shape shows the density of the property values.

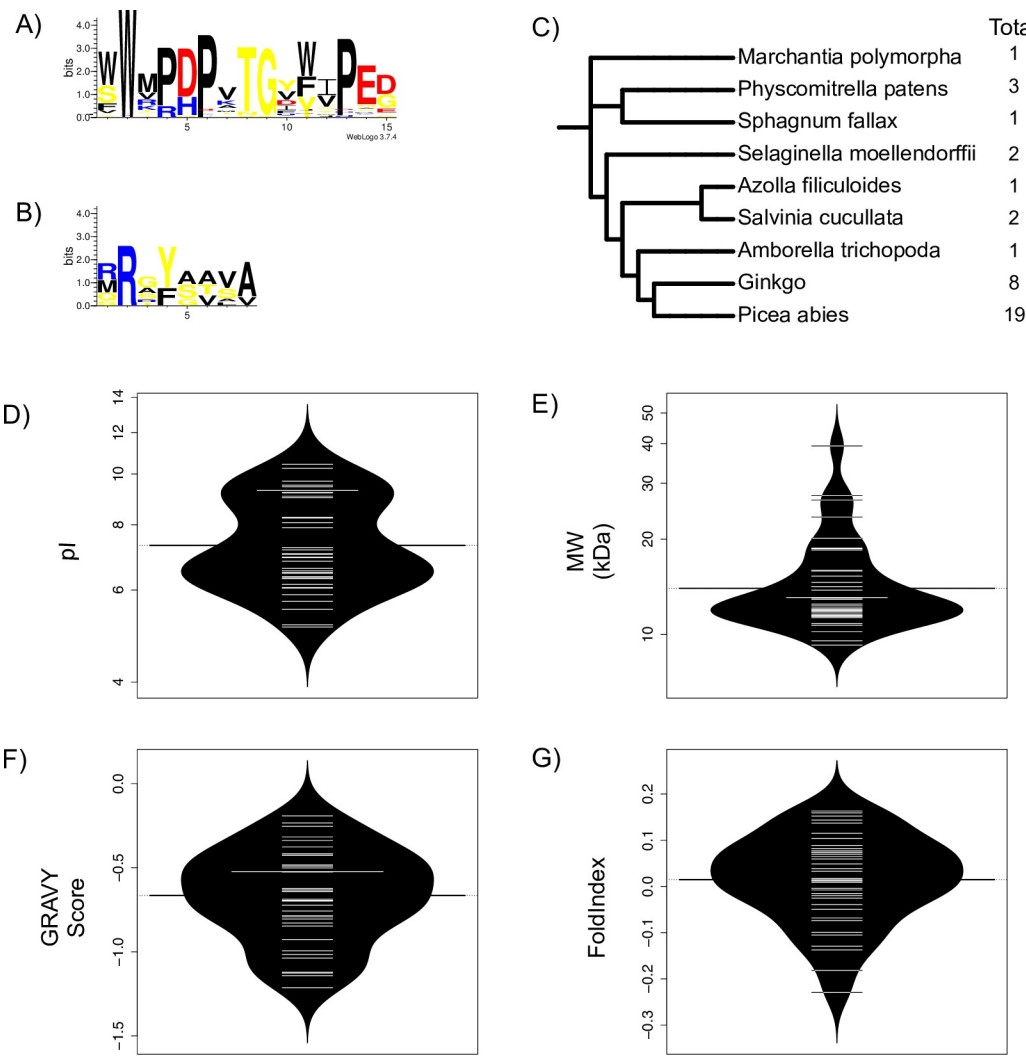

**Fig 5. LEA3 proteins in lower plants.** A) W-motif in lower plants in LOGO representation. B) RRGYA$_4$-motif in lower plants in LOGO representation. Amino acids are color-coded by their group type. Blue—positively charged (Lys, Arg, His); red—negatively charged (Asp, Glu); black—hydrophobic (Ala, Val, Leu, Ile, Pro, Phe, Met), yellow—polar (Gly, Ser, Thr, Tyr, Cys, Asn, Gln). The heights of the amino acids correspond to their level of conservation at that position. C) Species tree of LEA3 proteins in lower plants. The right-hand column shows the total number of proteins in each species. Bean plots of the D) isoelectric point (pI), E) molecular weight (MW), F) GRAVY score and G) FoldIndex score of LEA3 proteins in lower plants. The thin bars show the value of an individual protein, the wider black bar shows the mean value of an architecture, and the dotted line shows the mean value of all protein sequences. The violin shape shows the density of the property values.

with the only deviation being less conservation of the arginine in position 1, and a greater propensity for valine over alanine or serine to occur at position 7 (Fig 5B). We also looked at the number of LEA3 proteins in the lower plants (Fig 5C). For the most part, individual species had 1–3 proteins, with spermatophytes having considerably more at 8 (*Ginkgo biloba*) and 19 (*Picea abies*).

When comparing the physiochemical properties of the lower plants to the higher plants, we can see that the average pI (Fig 5D) for lower plants also ranges from pH 5.0–10.0, with an average pI of 7.4, making them slightly more acidic than the LEA3 proteins in higher plants. The MW of lower plant LEA proteins (Fig 5E) ranged from 9.3 kDa to 39.3 kDa, with an

average of 14.8 kDa. The GRAVY scores (Fig 5F) ranged from -1.2 to -0.19, with an average of -0.66, while the FoldIndex scores (Fig 5G) ranged from -0.23 to 0.16, with an average of 0.01.

## Evolution of LEA3 proteins

A phylogenetic tree was constructed using LEA3 protein sequences from both higher and lower plants (Fig 6), with the sequences labeled both by species name and by architecture. Although the bootstrap values near the middle of the tree are low, the clustering of the architectures often into single the same clade indicates that the evolutionary relationships can be analyzed. Firstly, the phylogenetic tree justifies the division of the different LEA3 proteins by the different architectures. The M[AS][RK] architecture formed only one clade, thought three MGRX sequences and one lower plant sequence were also found within this group. Likewise,

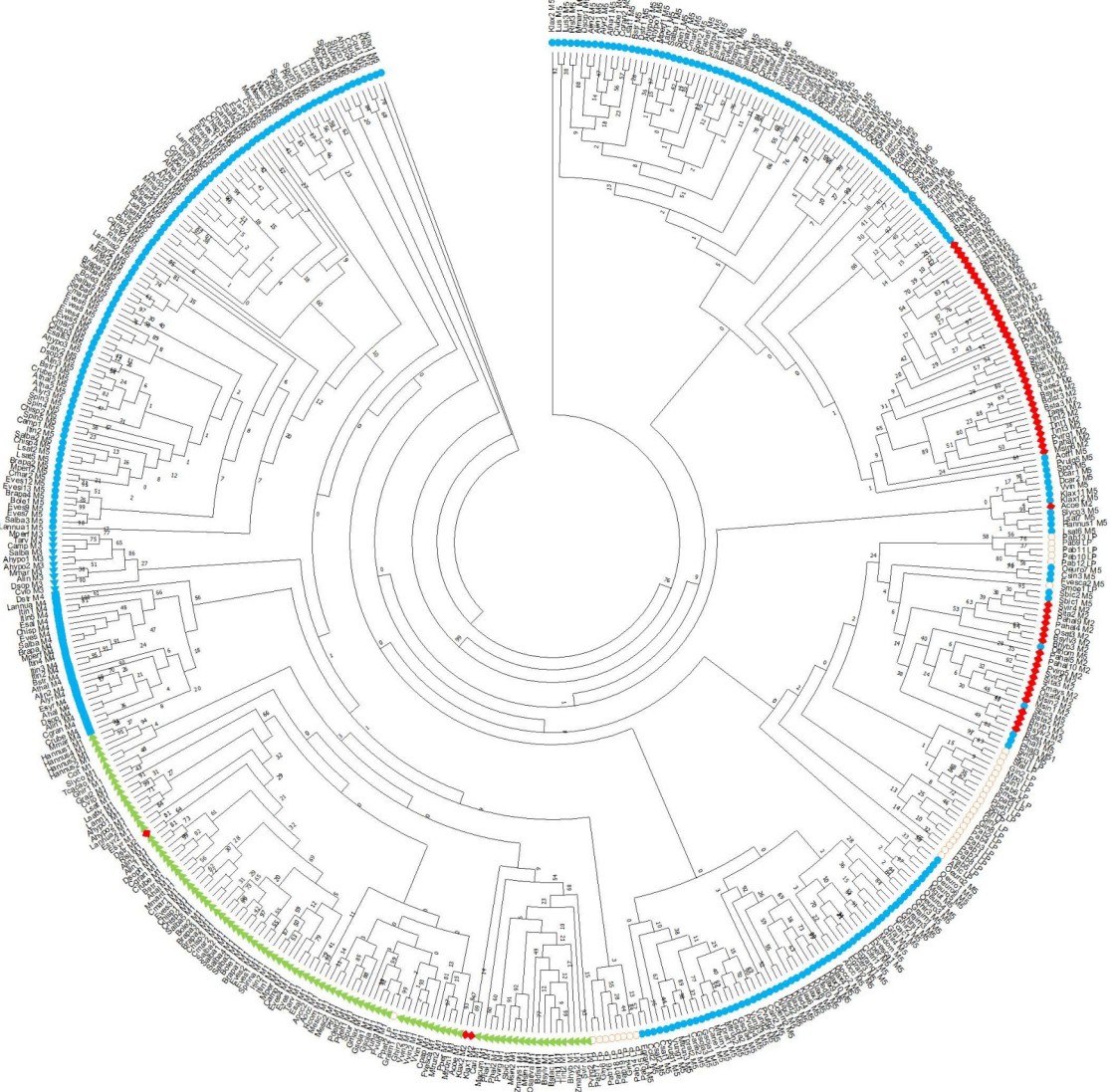

**Fig 6. Evolution of LEA3 genes.** Phylogenetic tree of LEA3 proteins from higher and lower plants. The tree was created using RAxML with 1000 bootstrap replicates. The bootstrap values are shown at each node. The architectures are coded as follows: MAaRS-1W, blue circles; MAaRS-2W, blue triangles; MAaRS-no DAELR, blue squares; MGRX, green triangles; M[AS][RK], red diamonds; Lower plants, open circles.

the MAaRS-W no DAELR architecture was located in adjacent clades to the MGRX architecture and the MAaRS-2W architecture was found in adjacent clades to the MAaRS-W no DAELR. Somewhat less consistency was observed for the MAaRS-1W and MGRX architecture in that they were not each found in a single clade; for MAaRS-1W, the sequences formed three large clades, while MGRX formed two. Lastly, the lower plants have a majority of species in one clade, with two other clades consisting dominantly of sequences from *Picea abies* and *Ginkgo biloba* LEA3 proteins.

## Discussion

### LEA3 protein sequences motifs

Our analysis of the protein sequences (primary transcripts) from the Phytozome v13 database [51] allowed us to identify 458 LEA3 proteins across a large diversity of higher plant species (S1 Fig). The use of a large number of species to search for LEA3 protein sequences provides us with an opportunity to determine which residues have been conserved over a long evolutionary time period, and hence provide key insight into LEA3 protein mechanisms in future biochemical studies. Analysis of the arrangements of the motifs, i.e. the protein architectures, can suggest that they may act as paralogs with different functions.

The first step was to identify the conserved motifs in this family of intrinsically disordered proteins. In addition to the previously identified W-motif, we also discovered two C-terminal motifs (named DAELR and EDVMP), four N-terminal motifs (named MAaRS, MARS, MGRX and M[AS][RK]) and observed the RRGYA$_4$ motif (Fig 1). The W-motif (Fig 1A) has been previously identified as a conserved motif in LEA3 proteins [11, 30–35]. A search of ELM motifs using TOMTOM from the MEME suite [52] did not reveal any similar motifs that have been previously described, and the lack of any identified biochemical function makes it currently challenging to propose anything beyond identifying the conserved residues.

The N-terminal motifs and the RRGYA$_4$ motif are likely to be the signals for the localization of LEA3 proteins inside the mitochondrion, which has been both predicted [27] and experimentally demonstrated for the *Arabidopsis thaliana* LEA3 proteins [27, 29]. Plant mitochondrial-targeting peptides typically possess a long stretch of amino acids with propensity to form an α-helix (i.e. the mitochondrial targeting sequence), followed by a putative cut site that has arginine at residues 2 & 3, or residues 3 & 4 (also known as the twin arginine motif), tyrosine or phenylalanine at -1, and alanine, serine, and threonine being common in the +1 and +2 positions [53, 54]. However, the arginine residues do not seem to be essential, since proteins lacking the arginine residues were also found to target to the mitochondria, having phenylalanine or tyrosine at the -1 and alanine or serine at the +1 position of the cut site. This may explain why some of the LEA3 proteins described here did not have an apparent twin-arginine motif, yet are still likely to locate to the mitochondrion. An additional reason may be that the N-terminal motif itself was captured as part of the twin-arginine motif. This is probably the case for the M[AS][RK] N-terminal motif, where the two C-terminal end residues are enriched in arginine (Fig 1D).

### LEA3 architectures

More insight can be found in the analysis of various LEA3 protein architectures (Fig 2A) and their distribution between (Fig 3A) and within (Fig 3B) plant species. As seen in Fig 2A, the LEA3 architectures can be defined by the presence and absence of different motifs. We observed that only a few proteins (~2.5%) lacked a conserved C-terminal motif, suggesting that these motifs must play an important role in the majority of LEA3 proteins. For LEA3 proteins with a C-terminal motif, the most common of these is the DAELR motif, which was

found in both MAaRS and M[AS][RK] architectures. The MGRX architecture is the only architecture to have the EDVMP C-terminal motif. While the two motifs are different (Fig 1B and 1C), they have some similarity in that both are rich in acidic amino acids, and have hydrophobic amino acids in the N- and C-termini of these motifs. We speculate that the C-terminal motifs may bind to different ligands, allowing the W-motif to bridge the same ligand bound by it to two different ligands bound by the two different C-terminal motifs. Confirmation of this proposal can come from more precise characterization of their location inside the mitochondrion.

Our motivation to examine the non-motif containing region of the LEA3 proteins (Fig 2B and 2C) comes from our work studying the sequence of another group of LEA proteins known as the dehydrins, where these regions are known as φ-segments. We counted the number of residues of this region in LEA3 proteins, which is situated between either the RRGYA$_4$ and W-motifs or the N-terminal and W-motifs for architectures lacking an apparent RRGYA$_4$ motif (Fig 2B). Interestingly, any difference in length seen between proteins containing or lacking the RRGYA$_4$ motif corresponds approximately to the length of the RRGYA$_4$ motif itself. The ~8 residue difference would further support the possibility of the RRGYA$_4$ motif being partially present in other N-terminal motifs, or being present in a slightly altered form in some of the sequences that could not be easily identified.

In the case of dehydrins, the φ-segment was found to be variably in length, ranging from 3 to 300 residues, with most being 30–50 residues long. In LEA3 proteins, the length of the non-conserved region was 10–50 residues, but was most commonly 27–37 residues, thereby being a smaller range than that of the dehydrins. When comparing overall amino acid composition of the variable regions between these two families of proteins, the most significant differences appear to be that alanine is more abundant (~15% in LEA3 vs ~7% in dehydrin), where the opposite is true for glycine (~10% in LEA3 vs ~17% in dehydrin). Likewise, a similar exchange occurs between threonine and serine, where serine is higher in LEA3 (~13% in LEA3 vs ~4% in dehydrin), but threonine is lower (~5% in LEA3 vs ~11% in dehydrin). The importance of this is unclear, since both glycine and alanine have similar disorder propensities, and both serine and threonine can be phosphorylated, though serine is generally more associated with disorder [55].

## Properties of LEA3 proteins

Previous analysis on biochemical properties of all LEA protein families has shown that they have a wide range of molecular weights (5–200 kDa), have pI values that are acidic, basic, or neutral, but are similar in that they are highly hydrophilic, and tend to have overrepresentation of glycine and underrepresentation of cysteine and aromatic amino acids, which explains why these proteins are mostly disordered [56, 57] When comparing LEA3 proteins to other members of the LEA protein family, the LEA3 average is on the lower end of the protein size range, with a value of 11.1 kDa. The average pI of LEA3 proteins is 9.0; however, the pI values of MAaRS and M[AS][RK] proteins are basic, while MGRX proteins have a group of proteins that is basic and another group that is acidic. With respect to the predicted structure, the high hydrophilic content is naturally reflected in both the GRAVY and FoldIndex scores. LEA3 proteins have an average GRAVY score of -0.44, which despite being negative, is higher (i.e. less hydrophilic) than what has been reported for most other LEA groups, which tend to cluster around -1.2 [57]. The FoldIndex scores correlated with the GRAVY scores, where the FoldIndex scores had an average value of +0.06. This value is slightly positive, whereas most other LEA proteins have scores that are on the negative side [57]. While FoldIndex scores close to zero suggest that the folding cannot be predicted confidently, preliminary biophysical results suggest that the *Arabidopsis thaliana* LEA3 proteins are disordered in solution.

## LEA3 evolution

Several analyses here provide information on the evolution of LEA3 proteins and its architectures (Figs 3 and 6 and S1 Fig). The most notable observation is that we could not find any LEA3 proteins in algae, despite using extensively broad searches with BLAST, FIMO and MAST with generous cut-offs, suggesting that the LEA3 proteins arose after the origin of land plants. This observation has also been made for LEA5 proteins (Pfam nomenclature, named LEA1 in the paper) [9] and also appears to be the case for dehydrins [36]. For LEA5 proteins, the authors argue that LEA proteins likely represent an important evolutionary event as plants moved from an aquatic environment to a terrestrial one [9]. This is further supported by the fact that the aquatic species *Zostera marina* does not have any LEA3 proteins that we could detect, though for this organism the absence likely represents the loss of the gene, an observation that has been made for other LEA genes in this species [58].

We also analyzed the distribution of the different architectures in higher plants to understand how they may have evolved (Fig 3). With respect to the MAaRS-2W architecture, we think that they represent an evolutionary variation of the MAaRS-1W protein that likely arose from a duplication of the W-motif, especially since they are concentrated in Malvids, though a small number of this architecture was found also in Fabids and Commelinids. Similarly, the M[AS][RK] architecture may represent an N-terminal motif variant of MAaRS-1W that is found in Commelinids (especially in grasses), based on both architectures having one W-motif and one DAELR motif (Fig 2A), and that both have similar values and ranges of their physiochemical properties (Fig 4).

An architecture that is different from the MAaRS/M[AS][RK] LEA3 proteins is the MGRX architecture. This is the only LEA3 protein group to contain the EDVMP motif (Fig 2A), and its physiochemical properties are different from the other two (Fig 4). Like the MAaRS-1W motif, MGRX proteins are present across all clades (Fig 3A). Note that the MGRX architecture was not detected in lower plants, suggesting that the MGRX motif arose early in the evolution of land plants.

## Supporting information

**S1 Fig. LEA3 proteins in higher plants by species.** A plant species tree was generated using PhyloT and NCBI genomic reference numbers as described in the Materials and Methods. The five architectures for each LEA3 protein are listed. The total for each species listed in the right-hand column, and the total of each architecture are listed in the bottom row.
(PDF)

**S1 Table. List of species used in this study.** Plant species names are divided into higher plants, lower plants, and algae.
(DOC)

**S1 Appendix. LEA3 protein sequences.** The file contains the sequences of all LEA3 proteins analyzed in this study in a FASTA file.
(FASTA)

## Author Contributions

**Conceptualization:** Steffen P. Graether.

**Data curation:** Karamjeet K. Singh.

**Formal analysis:** Karamjeet K. Singh, Steffen P. Graether.

**Supervision:** Steffen P. Graether.

**Visualization:** Karamjeet K. Singh, Steffen P. Graether.

**Writing – original draft:** Karamjeet K. Singh.

**Writing – review & editing:** Karamjeet K. Singh, Steffen P. Graether.

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
