## [Decision Letter · Decision Letter 0]

6 Jul 2020

PONE-D-20-14558

Conserved Sequence Motifs in the Abiotic Stress Response Protein Late Embryogenesis Abundant 3

PLOS ONE

Dear Dr. Graether,

Thank you for submitting your manuscript to PLOS ONE. After careful consideration, we feel that it has merit but does not fully meet PLOS ONE’s publication criteria as it currently stands. Therefore, we invite you to submit a revised version of the manuscript that addresses the points raised during the review process.

We look forward to receiving your revised manuscript.

Kind regards,

Ruslan Kalendar, PhD

Academic Editor

PLOS ONE

Journal Requirements:

Reviewers' comments:

Reviewer's Responses to Questions

**Comments to the Author**

1. Is the manuscript technically sound, and do the data support the conclusions?

 . 

Reviewer #1: Yes

2. Has the statistical analysis been performed appropriately and rigorously? 

Reviewer #1: N/A

3. Have the authors made all data underlying the findings in their manuscript fully available?

 Reviewer #1: Yes

4. Is the manuscript presented in an intelligible fashion and written in standard English?

Reviewer #1: Yes

5. Review Comments to the Author

Reviewer #1: 

The paper focused on the characterisation and sequencing of LEA3 proteins. For this, the authors performed bioinformatics analyses of these dehydrins using Phytozome database and Pfam LEA3 sequences as sources ultimately creating a LEA3 sequence dataset. The authors separated the data for vascular and non-vascular plants and studied the spread of LEA3 proteins in plant spp.

1. The authors should think about Figs 1, 2A, 5A, 5B in terms of accessibility particularly when accessed by colour blind readers. The figures mentioned have the potential to be completely uninformative if viewed in wrong conditions.

2. I couldn’t access S1 Figure but I think it would be great to have principal components showing some sort of clustering of the LEA3 proteins identified with respect to the plant species

3. Line 469 This is not true. There are reviews and articles that showed that the molecular weight of LEA proteins could be up to 200KDa. See:

a) Hanin, M., Brini, F., Ebel, C., Toda, Y., Takeda, S. and Masmoudi, K., 2011. Plant dehydrins and stress tolerance: versatile proteins for complex mechanisms. Plant signaling & behavior, 6(10), pp.1503-1509;

b) Graether, S.P. and Boddington, K.F., 2014. Disorder and function: a review of the dehydrin protein family. Frontiers in plant science, 5, p.576;

c) Ogbaga, C.C., Stepien, P., Dyson, B.C., Rattray, N.J., Ellis, D.I., Goodacre, R. and Johnson, G.N., 2016. Biochemical analyses of sorghum varieties reveal differential responses to drought. PloS one, 11(5).

d) Yu, Z., Wang, X. and Zhang, L., 2018. Structural and functional dynamics of dehydrins: a plant protector protein under abiotic stress. International journal of molecular sciences, 19(11), p.3420.

Language

Line 15 change containing to contain

Line 25 change arose to arisen

6. PLOS authors have the option to publish the peer review history of their article (what does this mean?). If published, this will include your full peer review and any attached files.

Reviewer #1: **Yes: **Chukwuma C. Ogbaga

---

## [Author Response · Author response to Decision Letter 0]

19 Jul 2020

Dear Editor,

Thank-you for the opportunity to submit a revision of our manuscript titled “Conserved Sequence Motifs in the Abiotic Stress Response Protein Late Embryogenesis Abundant 3.” We hope that addressing these issues will allow our manuscript to be published.

Response to the Editor

1) We have revised our manuscript to ensure that it follows the PLOS One formatting guidelines.

2) We have removed the “data not shown” statements and rewritten the text to indicate that a) no algae sequences were found so that there is no data shown and b) the data for the lower plant N-terminal sequences are shown in the supplemental data.

Response to Reviewer #1

1) We thank the reviewer for pointing out that our figures did not take color vision deficiency into account. We have corrected the Figures 1, 2, and 5 to ensure that the colors and color shades are appropriate now.

2) The suggestion of using PCA to see if there is any clustering of sequences by species is interesting, but one that we feel that the reviewer would not see as necessary if they were able to open Fig. S1. There does not seem to be any significant trend, other than our previously described observation that the M[AS][RK] N-terminal sequence is predominantly found in Commelinids. We are also unclear on what variables or categories would be used for the PCA. We therefore felt that adding a PCA analysis would not significantly improve the manuscript.

3) We corrected the upper MW limit of dehydrins (now Line 566 in the revised with track-changes manuscript).

4) We corrected ‘containing’ to ‘contain’ (now line 18) and ‘arose’ to ‘arisen’ (now line 28).

Sincerely,

Steffen Graether, Ph.D.

Associate Professor

Department of Molecular and Cellular Biology

University of Guelph

Guelph, Ontario

---

## [Editor Report · Decision Letter 1]

22 Jul 2020

Conserved Sequence Motifs in the Abiotic Stress Response Protein Late Embryogenesis Abundant 3

PONE-D-20-14558R1

Dear Dr. Graether,

We’re pleased to inform you that your manuscript has been judged scientifically suitable for publication and will be formally accepted for publication once it meets all outstanding technical requirements.

Kind regards,

Ruslan Kalendar, PhD

Academic Editor

PLOS ONE

---

## [Editor Report · Acceptance letter]

27 Jul 2020

PONE-D-20-14558R1 

Conserved sequence motifs in the abiotic stress response protein late embryogenesis abundant 3 

Dear Dr. Graether:

I'm pleased to inform you that your manuscript has been deemed suitable for publication in PLOS ONE. Congratulations! Your manuscript is now with our production department. 

Kind regards, 

on behalf of

Dr. Ruslan Kalendar 

Academic Editor

PLOS ONE